# Learning Neural Lyapunov Functions to Characterize Stability Regions for Unknown Dynamical Systems

## Abstract

Lyapunov function is often used as a mathematical tool to evaluate the stability of dynamical systems by demonstrating that system trajectories converge to an equilibrium point. This work develops a machine learning method for designing Lyapunov functions and characterizing stability regions in nonlinear dynamical systems with unknown first-principles models. The Lyapunov function is developed as a neural network model with its architecture and loss function designed to ensure that the conditions of a control Lyapunov function are satisfied. The optimal Lyapunov neural network is identified using Bayesian optimization that maximizes the estimated stability region. Theoretical guarantees are provided to ensure that, despite approximation errors, the Lyapunov function and the stability region derived from the data remain valid for the underlying nonlinear system. The proposed method is applied to various nonlinear systems, demonstrating its effectiveness in Lyapunov function design and stability region characterization.

## 1 Introduction

Lyapunov stability theory is a fundamental tool for analyzing the stabilization of nonlinear systems and has attracted significant attention from researchers and engineers (Chang et al., 2019; Debauche et al., 2024; Wu et al., 2023; Liu et al., 2024). It has led to numerous theoretical advancements and successful applications across various engineering fields (Khalil, 2002; Kang et al., 2022; Mirzaei & Mathis, 2024). The stability region (also known as the region of attraction) is an invariant set determined by the Lyapunov function and plays a crucial role in system operation and controller design. If the initial state lies within this region, the system's trajectory remains inside and ultimately converges to the steady state. Several methods have been proposed to design the Lyapunov function and stability region, including sum-of-squares (SOS) method (Papachristodoulou & Prajna, 2002) and Zubov's equation (Kang et al., 2023). Despite these advances, designing Lyapunov functions that provide large and accurate stability regions for nonlinear systems with unknown first-principles models remains a challenging and open research problem.

With the development of computational resources, machine learning algorithms have increasingly been applied to the design of Lyapunov functions, with Lyapunov neural network (LNN) methods receiving significant attention (Alfarano et al., 2024; Debauche et al., 2024; Jena et al., 2024; Chang et al., 2019; Zhou et al., 2022; Grüne, 2020; Zhai & Nguyen, 2021; Min et al., 2023). Unlike conventional polynomial Lyapunov functions, LNNs represent Lyapunov functions as neural networks, where parameters are updated via a designed loss function. For example, in Chang et al. (2019), the control policy and the neural Lyapunov function were learned with provable guarantees of stability via the proposed framework consisting of a learner and a falsifier. The unknown nonlinear dynamics, controller, and Lyapunov function were learned simultaneously in Zhou et al. (2022). Additionally, the design of neural Lyapunov functions was investigated in Liu et al. (2025) via physics-informed neural networks and Zubov's equation.

Despite these recent efforts, existing methods for learning Lyapunov functions via neural networks face several key challenges (Dai et al., 2021; Chang et al., 2019; Zhou et al., 2022; Abate et al., 2020; Richards et al., 2018): (i) ensuring that the learned function is bounded by class $\mathcal{K}$ functions / quadratic functions, which is required by the converse Lyapunov theorem to design Lyapunov

functions that can rigorously prove and quantify asymptotic / exponential stability, (ii), deriving theoretical guarantees of system stability using the Lyapunov function constructed from data, and (iii) designing a Lyapunov function typically requires a known system model, which is often unavailable or incomplete in real-world applications. For example, in Chang et al. (2019), the loss function was designed to quantify the violation of Lyapunov conditions, and its value decreased during training, guiding the model toward satisfying stability criteria. Various neural network structures were proposed and utilized to perform the Lyapunov function (Samanipour & Poonawala, 2023; Yang et al., 2024; Zhang et al., 2022; Kolter & Manek, 2019). In Gaby et al. (2022), a neural network structure was designed to ensure the positive definiteness of the Lyapunov neural network function. While recent efforts have introduced various approaches for designing Lyapunov functions, they did not fully address the aforementioned challenges. Furthermore, most existing studies on Lyapunov function design and stability region characterization assume the availability of an accurate first-principles model for the nonlinear system. However, in practice, developing such a model is often challenging due to the presence of complex physicochemical interactions that are difficult to capture. Therefore, despite the progress made in the previous research, developing a data-driven framework that simultaneously learns system dynamics and constructs Lyapunov functions with theoretical stability guarantees remains an unresolved problem in the field.

In addition to the design of LNNs that satisfy the conditions of Lyapunov functions, identifying an optimal LNN that maximizes the stability region for a nonlinear system remains a critical challenge of practical importance. An improperly designed Lyapunov function can result in an overly conservative estimate of the stability region, which not only restricts the allowable operating range of real-world systems under stability guarantees, but may also hinder performance, reduce efficiency, and limit the system's ability to adapt to dynamic environments or disturbances. Several methods have been proposed to enhance LNN performance (Richards et al., 2018; Chang et al., 2019; Dai et al., 2021; Wang & Wu, 2024; Polyak & Shcherbakov, 2017). For example, in Chang et al. (2019), a learner-falsifier framework was introduced to iteratively exclude states that violate Lyapunov conditions. The neural network Lyapunov function and neural-network controller were synthesized via a min-max optimization problem in Dai et al. (2021). In Polyak & Shcherbakov (2017), the design of the Lyapunov function was considered from an optimization perspective. Inspired by the previous results, we aim to design a Bayesian optimization framework to select the optimal LNN within a given class of hypothesis functions (Frazier, 2018; Smith, 1987).

Motivated by the above considerations, this work investigates the data-driven design of LNNs and the estimation of the stability region for the nonlinear system with unknown process models. A novel neural network architecture is proposed for the design of Lyapunov functions, and optimization problems are formulated to identify the optimal LNN that maximizes the stability region. Theoretical guarantees are provided to ensure that the Lyapunov function and the stability region derived from data remain valid for the nonlinear system.

## 2 PRELIMINARIES

### 2.1 DEFINITIONS AND NOTATIONS

$\|x\|$ denotes the Euclidean norm of a vector $x \in \mathbb{R}^n$. $x^T$ denotes the transpose of $x$. A function $f : \mathbb{R}^n \to \mathbb{R}^m$ is said to be Lipschitz continuous, if there exists a constant $L > 0$ such that $\|f(x) - f(y)\| \leq L\|x - y\|$ for all $x, y \in \mathbb{R}^n$. A continuous function $\beta : [0, b) \to [0, \infty)$ is of class $\mathcal{K}$ if it is strictly increasing and $\beta(0) = 0$.

### 2.2 CLASS OF NONLINEAR SYSTEMS

In this work, a class of nonlinear systems described by the following ordinary differential equation (ODE) is considered:
$$\dot{x} = F(x, u) \tag{1}$$
where $x \in \mathbb{R}^n$ and $u \in \mathbb{R}^m$ are the state and input vectors, respectively. $\dot{x}$ denotes the time derivative of the state vector $x$. $F$ is a smooth nonlinear function. The operating region of the state vector $D$ is a compact subset of $\mathbb{R}^n$. The input vector is bounded by $u \in U = \{u_i^l \leq u_i \leq u_i^u, i = 1, \dots, m\}$. We assume that $F(0, 0) = 0$, and thus, the origin $(x, u) = (0, 0) \in \mathbb{R}^n \times \mathbb{R}^m$ is a steady-state of the nonlinear system of Eq. 1.

In this work, we assume that there exists a stabilizing controller $u = \Phi(x) \in U$ under which the origin of the nonlinear system in Eq. 1 is rendered exponentially stable. According to the converse Lyapunov theorem (Khalil, 2002), it implies there exists a control Lyapunov function $V(x)$ that satisfies the following inequalities for all $x$ within the region $D$ ($D$ is a neighborhood around the origin):

$$c_1 \|x\|^2 \leq V(x) \leq c_2 \|x\|^2 \tag{2a}$$

$$\frac{\partial V(x)}{\partial x} F(x, \Phi(x)) \leq -c_3 \|x\|^2 \tag{2b}$$

$$\left\| \frac{\partial V(x)}{\partial x} \right\| \leq c_4 \|x\| \tag{2c}$$

where $c_1$, $c_2$, $c_3$, and $c_4$ are positive constants. To characterize the stability region for the nonlinear system of Eq. 1, we first identify a region where the time derivative of $V$ is rendered nonpositive as follows: $\Omega_d = \{x \in \mathbb{R}^n \mid \frac{\partial V(x)}{\partial x} F(x, \Phi(x)) \leq -kV\}$, where $k$ is a positive constant. Then, the stability region can be defined as a level set of the Lyapunov function within $\Omega_d$: $\Omega_\rho = \{x \in \Omega_d \mid V(x) \leq \rho\}$. Note that for any initial state $x_0$ within the region $\Omega_\rho$, we have $x(t) \in \Omega_\rho$ when $t > 0$. Since $F(x, u)$ is assumed to be a Lipschitz continuous function with respect to $x$ and $u$, the following inequalities hold for $x, x' \in D$ and $u, u' \in U$.

$$\|F(x, u)\| \leq B_F \tag{3a}$$

$$\|F(x, u) - F(x', u')\| \leq L_1 \big( \|x - x'\| + \|u - u'\| \big) \tag{3b}$$

$$\|\nabla V(x)F(x, u) - \nabla V(x')F(x', u')\| \leq L_2 \big( \|x - x'\| + \|u - u'\| \big) \tag{3c}$$

where $B_F$, $L_1$, and $L_2$ are positive constants.

This work develops a machine learning-based method for the design of a Lyapunov function that maximizes the stability region for the nonlinear system of Eq. 1, where an accurate process model is unavailable. A neural network model $F_{nn}(x, u)$ is first constructed from data to capture the dynamics of the nonlinear system. However, while the Lyapunov function and the stability region are designed using data, we prove that they remain valid for the actual nonlinear system in the presence of approximation errors. Note that the size of the stability region varies depending on the choice of Lyapunov function and the stabilizing controller. In this manuscript, we focus on the design of Lyapunov functions that maximize the stability region, and do not consider the effect of the stabilizing controller (i.e., a stabilizing controller is chosen and remains unchanged during the optimization process).

## 3 DESIGN OF LNN

In this section, we design the Lyapunov function in the form of a feedforward neural network (FNN) model. A theoretical analysis is provided to demonstrate that the LNN satisfies the conditions of a Lyapunov function. Furthermore, the stability region obtained for the neural network model remains valid for the actual nonlinear system in Eq. 1.

A polynomial function with quadratic form can be chosen as a Lyapunov candidate because it meets the fundamental requirements of a Lyapunov function. However, a standard neural network model does not inherently satisfy the requirements for the Lyapunov function in Eq. 2a and Eq. 2c. Therefore, structural modifications are necessary to ensure that the neural network can function as a valid Lyapunov function.

First, we develop an FNN model that satisfies Eq. 2a and Eq. 2c. To simplify the discussion, we consider an FNN model with one hidden layer of $m$ neurons and one output layer. Specifically, the shape of the NN input is $(p, n)$, where $p$ and $n$ are the number of samples and the state variables, respectively. The network output represents the Lyapunov function values for the $p$ corresponding state sets. For $x = [x_1, \ldots, x_n]^T \in D$, the output of the FNN can be calculated as follows:

$$\mathbf{h}_j = h_j(\sum_{i=1}^{n} w_{ji} x_i + b_j), \; for \; j = 1, \ldots, m; \; \mathbf{o} = o(\sum_{j=1}^{m} w_{oj} h_j + b_o) \tag{4}$$

where $\mathbf{h}_j$ is the hidden state of the $j$th neuron, and $\mathbf{o}$ is the output of the neural network. $w$ and $b$ are the weight parameters and the bias in the hidden and output layers, respectively. $h$ and $o$ are the activation functions for the hidden layer and the output layer. Additionally, an adaptation layer $\phi(x)$ is added to fulfill the requirements of the Lyapunov function. With this modification, the final output of the neural network can be computed as follows:

$$V_{nn}(x) = o(\sum_{j=1}^{m} w_{oj} h_j(\sum_{i=1}^{n} w_{ji} x_i + b_j) + b_o) + \phi(x) \tag{5}$$

**Theorem 1.** *Consider the FNN in Eq. 5, for any $x = [x_1, \ldots, x_n]^T \in D$. If the neural network structure, weight parameters, and the adaptation term satisfy*

$$b_j = 0, \quad b_o = 0, \quad 0 < LB_w \le w_{ji} \le UB_w \text{ for } j = 1, \ldots, m \text{ and } i = 1, \ldots, n \tag{6a}$$

$$h(t) \le k_1 |t|, \quad 0 < \frac{\mathrm{d}h(t)}{\mathrm{d}t} \le k_2 \tag{6b}$$

$$0 \le o(t) \le k_3 t^2, \quad \left| \frac{\mathrm{d}o(t)}{\mathrm{d}t} \right| \le k_4 t \tag{6c}$$

$$k_5 \|x\|^2 \le \phi(x) \le k_6 \|x\|^2, \quad \left\| \frac{\mathrm{d}\phi(x)}{\mathrm{d}x} \right\| \le k_7 \|x\| \tag{6d}$$

*where $k_1$, $k_2$, $k_3$, $k_4$, $k_5$, $k_6$, and $k_7$ are all positive constants. $h(t)$ and $o(t)$ are element-wise functions chosen as the activation function and $\phi(x)$ denotes the adaptation neural network layers, then the following inequality holds $\forall x \in D$:*

$$c_1 \|x\|^2 \le V_{nn}(x) \le c_2 \|x\|^2 \tag{7a}$$

$$\left\| \frac{\partial V_{nn}(x)}{\partial x} \right\| \le c_4 \|x\| \tag{7b}$$

*where $0 < c_1 \le k_5$, $2k_3(mnk_1 UB_w^2)^2 + k_6 \le c_2$, and $k_1 k_2 k_4 m^2 n^2 UB_w^4 \sqrt{n} + k_7 \le c_4$.*

The proof of Theorem 1 can be found in Appendix A.1. Theorem 1 shows that by properly designing the FNN, it can be a valid Lyapunov function in the sense that the conditions in Eq. 2 are satisfied. Additionally, the conditions required in Theorem 1 are also practically achievable. For example, the boundness of the weight parameters in Eq. 6a can be met using the 'clamp' function in Pytorch. The activation functions for the proposed LNN can be selected to meet the conditions in Eq. 6b and Eq. 6c. For example, the activation function $h(t)$ in the hidden layer can be chosen as $t$ and $tanh(t)$. The activation function $o(t)$ can be chosen as $kt^2$ to meet the requirement in Eq. 6c, where $k$ is a positive constant. The discussion of Theorem 1 can be found in Appendix B.1.

Following the design of network architecture, the loss function of the LNN is designed as follows:

$$L = \min_{w_j} Relu(\frac{\partial V_{nn}(x)}{\partial x} F_{nn}(x, \Phi_{nn}(x)) + \lambda V_{nn}(x)) \tag{8}$$

where $\Phi_{nn}(x)$ is the neural network-based stabilizing controller. $w_j$ is the weight parameters in the designed LNN. $\lambda$ is a positive constant. $Relu$ is the nonlinear function that outputs the input if it is positive; otherwise, the output is zero.

To obtain the LNN and the region $D$, where Eq. 2 holds for the proposed LNN within $D$, a new region is designed as $D^L$, where $D \subseteq D^L$. A dataset $\mathcal{S}$ of $s$ samples drawn independent and identically distributed (i.i.d.) from $D^L$ is collected. Then, the loss function is calculated for all samples within $\mathcal{S}$, with the goal of ensuring that the LNN is effective on the data samples. Specifically, the time derivative of the LNN function $\frac{\partial V_{nn}(x)}{\partial x} F_{nn}(x, \Phi_{nn}(x))$ should be less than $-\lambda V_{nn}(x)$. Since the output of the $Relu$ function is zero if the input is negative, the requirements of the Lyapunov function hold for $\mathcal{S}$, when the loss of the LNN model is reduced to $0$ in the data set $\mathcal{S}$.

Since the LNN is trained with a dataset of finite state samples, the analysis of the generalization performance for all states in the state-space is important for the implementation of LNN to real-world systems. Theorem 2 shows that if Eq. 2b holds for the data sample $x_p$ in the training set, all the states within the neighborhood around $x_p$ can be stabilized via the same candidate controller. The proof of Theorem 2 can be found in Appendix A.2. Subsequently, Corollary 1 provides guidance on how to select training samples to ensure that the conditions of the Lyapunov function are satisfied for all states beyond the training samples.

**Theorem 2.** *Consider the nonlinear system $\dot{x} = F(x, u)$ of Eq. 1, and the LNN $V_{nn}$ designed following the conditions in Theorem 1 and the loss function in Eq. 8. If there exists $c_a > L_2 > c_b > 0$, where $L_2$ is the Lipschitz constant for $\frac{\partial V_{nn}(x)}{\partial x} F(x, \Phi(x))$ in Eq. 3, such that for any data sample $x_p \in \mathcal{S}$, the following condition holds:*

$$\frac{\partial V_{nn}(x_p)}{\partial x} F(x_p, \Phi(x_p)) \leq -c_a \left\| x_p \right\|^2 \tag{9}$$

*then the following inequality holds $\forall x \in \mathcal{B}_r(x_p, r)$:*

$$\frac{\partial V_{nn}(x)}{\partial x} F(x, \Phi(x)) \leq -c_b \left\| x \right\|^2 \tag{10}$$

*where $\mathcal{B}_r(x_p, r)$ is the neighborhood of the data point $x_p$ defined as $\mathcal{B}_r(x_p, r) = \{x \in D \mid \|x - x_p\| \leq r\}$, and $0 < r \leq \frac{c_a - c_b}{L_2 - c_b} \left\| x_p \right\|^2$.*

**Corollary 1.** *If the training samples $x_p \in \mathcal{S}$ are evenly distributed and satisfy Eq. 9, and the minimum distance between two training samples satisfies $d_p = \min\limits_{x \in \mathcal{S}} \|x - x_p\| \leq \frac{c_a - c_b}{L_2 - c_b} \left\| x_p \right\|^2$, $\forall x_p \in \mathcal{S}$, then a closed region can be constructed as $\Omega_S = \bigcup_{x_p \in \mathcal{S}} \mathcal{B}_r(x_p, d_p)$ such that all states within $\Omega_S$ satisfy the Lyapunov stability condition of Eq. 10.*

It is noted that the design and evaluation of Lyapunov function requires a process model $F(x, u)$. However, in this article, the first-principles model in Eq. 1 is assumed to be unavailable for the design of Lyapunov functions. Therefore, a data-driven model in the form of an FNN is constructed to approximate the system dynamics. The model is then used in the design of the controller and the LNN. The existence of approximation errors must also be taken into account to ensure that the stability region obtained using the FNN model remains effective for the nonlinear system of Eq. 1. Specifically, the FNN model that captures the system dynamics is developed in the following form:

$$\dot{\hat{x}} = F_{nn}(\hat{x}, u) \tag{11}$$

where $\hat{x} \in \mathbb{R}^n$ is the predicted state of the FNN model, and $u \in \mathbb{R}^m$ is the control action.

**Theorem 3.** *Consider the nonlinear system of Eq. 1 associated with the FNN in Eq. 11 that captures the nonlinear dynamics and an LNN developed following the conditions in Theorem 1 and the loss function in Eq. 8. If there exists a constant $\lambda \geq \frac{C_\lambda P_V}{k_5} > 0$, an FNN model that satisfies the modeling error constraint $\|v\| = \|F(x, u) - F_{nn}(x, u)\| \leq C_\lambda \|x\|$ for all $x \in \mathcal{S}$ and $u \in U$, where $C_\lambda$ is a positive constant and $P_V = k_1 k_2 k_4 m^2 n^2 U B_w^4 \sqrt{n} + k_7$, such that for any data sample $x \in \mathcal{S}$ that satisfies:*

$$\frac{\partial V_{nn}(x)}{\partial x} F_{nn}(x, \Phi_{nn}(x)) \leq -\lambda V_{nn}(x) \tag{12}$$

*Then, the following inequality holds:*

$$\frac{\partial V_{nn}(x)}{\partial x} F(x, \Phi_{nn}(x)) \leq -c_d V_{nn}(x) \tag{13}$$

*where $0 \leq c_d \leq \frac{\lambda k_5 - C_\lambda P_V}{2 k_3 (m n k_1 U B_w^2)^2 + k_6}$.*

The proof of Theorem 3 can be found in Appendix A.3. Theorem 3 shows that under a proper design of the LNN, the stability region designed for the neural network is effective for the actual nonlinear system, despite the approximation error between $F_{nn}(x, u)$ in Eq. 11 and the actual nonlinear system $F(x, u)$. Specifically, the parameters in the loss function should be designed based on the modeling error bound parameter ($C_\lambda$), the bound of the partial derivative of the LNN with respect to the state variable ($P_V$), and the regularization term included in the LNN ($k_5$). The discussion of Theorem 3 can be found in Appendix B.2.

## 4  DESIGN OF THE OPTIMAL STABILITY REGION

In this section, an optimization problem is formulated to find the optimal LNN model that produces the largest stability region in a data-based context. Specifically, the stability region is first represented by a set of state samples, for which the size is measured by the number of states inside the

set. Subsequently, the Bayesian optimization method is utilized to find the optimal LNN within a set of hypothesis functions that leads to the largest stability region. Note that traditionally the stability region is represented by the level set $V \leq \rho$, where $\rho$ indicates its size. However, in this work, the Lyapunov function is developed as a neural network, and therefore $\rho$ varies with network parameters and is not a reliable size metric across different Lyapunov functions. We instead quantify the region by the number of state samples it contains, i.e., more samples indicating a larger region.

Specifically, if a Lyapunov function has been designed (e.g., $V_{nn}$), the largest stability region for the neural network model $F_{nn}$ can be obtained via solving the following optimization problem:

$$\mathcal{J}_\rho = \max_\rho \{N_\rho\} \tag{14a}$$

$$\text{s.t.} \quad x \in \mathcal{S} \tag{14b}$$

$$\frac{\partial V_{nn}(x)}{\partial x} F_{nn}(x, \Phi_{nn}(x)) \leq -\lambda V_{nn}(x), \quad \text{if } V_{nn}(x) \leq \rho \tag{14c}$$

$$min(V_{nn}(x)) \leq \rho \leq max(V_{nn}(x)) \tag{14d}$$

where $N_\rho$ denotes the number of state samples that satisfy Eqs. 14b-14d. Eq. 14b ensures that the state $x$ is within the dataset $\mathcal{S}$. Eq. 14c is designed based on the definition of the stability region, that is, the time derivative of the Lyapunov function $V_{nn}$ under the neural network-based controller $\Phi_{nn}$ is negative for the state within a region $V_{nn}(x) \leq \rho$ for the neural network model $F_{nn}$. Moreover, $\lambda$ in Eq. 14c aims to ensure that the stability region obtained for the neural network model remains valid for the actual nonlinear system based on Theorem 3.

The optimized variable is selected as $\rho$, of which the value is within the region $[min(V_{nn}(x)), max(V_{nn}(x))]$ for $x \in \mathcal{S}$. Due to the complexity in calculating the time derivative of the LNN model in Eq. 14c, the solution of this optimization problem is not trivial. Moreover, Eq. 14 serves as a basis for further optimization of the LNN model to identify the largest stability region. Therefore, the global search method is used to find the optimal $\rho$ within the region $[min(V_{nn}(x)), max(V_{nn}(x))]$. This approach ensures that the stability region containing the largest number of samples is identified, and the global optimal solution is guaranteed.

Characterizing the stability region by the number of states in the level set enables a data-driven comparison of region sizes across different Lyapunov functions. Although Eq. 14 can be evaluated once a Lyapunov function is given, different functions may yield vastly different stability regions. Therefore, identifying the largest stability region requires explicitly accounting for this objective during the Lyapunov function design process. Specifically, the objective of this work is to design a Lyapunov function that not only satisfies the conditions of Eq. 2 but also maximizes the stability region. To this end, the following optimization problem is developed to obtain the optimal LNN from a hypothesis class:

$$\mathcal{J} = \max_{V_{nn} \in \mathcal{H}} \{\max_\rho \{N_\rho\}\} \tag{15}$$

where $\mathcal{H}$ is the hypothesis set of neural network functions. In the optimization process for Eq. 15, the hypothesis set $\mathcal{H}$ is constructed as a set of neural network functions that satisfy the requirements in Theorem 1 and are trained under the loss function of Eq. 8. Then, the optimized variables are selected, which can be the weight boundaries for the LNN model, the training epochs, and the number of neurons. The objective function of Eq. 15 captures the relationship between the set of optimized variables and the number of state samples within the stability region, which is approximated by the Gaussian process (GP) regression function. At each iteration, a new value set of the optimized variables is selected via the expected improvement (EI) acquisition function, which is calculated using the surrogate function. For the selected LNN model, the largest stability region is identified with the global search method by solving the optimization problem in Eq. 14, and the number of state samples within the stability region is the value of the loss function. The surrogate function is updated with the selected samples after each iteration, accelerating the optimization process since the previous information is utilized. The optimization process ends when a predefined criterion is satisfied (e.g., the value of the objective function is not increased for several iterations).

*Remark* 1. The major contributions of the proposed data-driven framework are as follows: 1) a novel structure to design Lyapunov neural networks and maximize the stability region. 2) theoretical analysis to ensure the Lyapunov stability conditions and the validity of the stability region. 3) Bayesian optimization problem formulated to enlarge the stability region. The advantages of the proposed framework compared to the existing results are discussed in detail in the Appendix B.3 and validated via the simulation results in the later Section.

## 5 CASE STUDIES

In this section, the effectiveness of the proposed data-driven framework for designing LNN functions and stability regions is demonstrated. Larger stability regions are identified using the proposed method for various nonlinear systems compared to existing methods. Specifically, the baseline includes the sum-of-squares (SOS) (Papachristodoulou & Prajna, 2005; Peet & Papachristodoulou, 2012; August & Barahona, 2022), the polynomial function (PF) (Feng et al., 2024; Wu et al., 2019), and the neural Lyapunov function (NLF) (Zhou et al., 2022; Chang et al., 2019; Min et al., 2023). In all case studies, the Adam optimizer is utilized to update the weight parameters, and the optimal LNN is obtained by the Bayesian optimization method. The training and optimization processes are conducted on Intel Core i7 with 32 GB of RAM.

### 5.1 VAN DER POL OSCILLATOR

We begin by applying the proposed LNN to a nonlinear system to illustrate the effectiveness of the method in finding the optimal Lyapunov function that maximizes the stability region. The Van der Pol oscillator is a classic nonlinear system without input and is characterized by two state variables. SOS (Peet & Papachristodoulou, 2012; August & Barahona, 2022), NLF (Zhou et al., 2022), and PF (Feng et al., 2024) serve as baselines. LNN 1, SOS 1, SOS 2, and PF are developed under the assumption that the system dynamics are known, whereas NLF and LNN 2 are trained without access to the true dynamics. Specifically, LNN 2 shares the same structure as LNN 1 but employs a feedforward neural network (FNN) to approximate the system dynamics with sufficient accuracy.

Both LNN models satisfy the conditions for the Lyapunov function defined in Eq. 2a and Eq. 2c. Although the stability region identified using LNN 2 is slightly smaller than that of LNN 1, it remains valid for the actual nonlinear system and is still significantly larger than those obtained by SOS 1, NLF, and PF, as shown in Fig. 1. In all the baselines, only SOS 2 achieves a slightly larger stability region than LNNs. However, the ability of LNNs to handle non-polynomial nonlinear functions and to flexibly address unknown nonlinear dynamical systems makes them a more general and effective choice compared to SOS. Note that LNN 1 and LNN 2 achieve stability regions approximately $100\%$ larger than those achieved by PF and NLF. A detailed description of the Van der Pol oscillator and simulation results can be found in Appendix C.

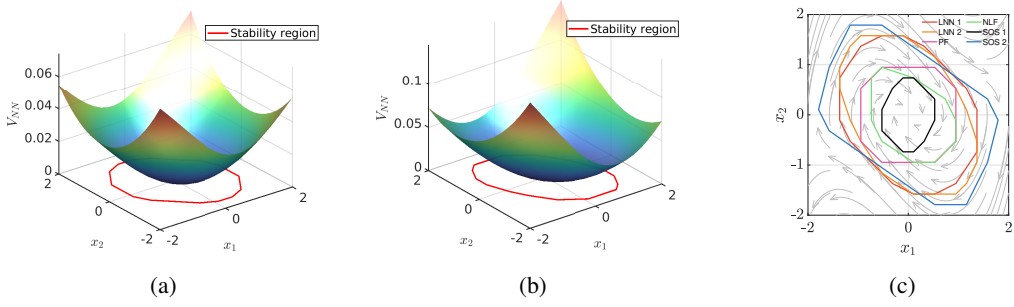

(a)           (b)           (c)

Figure 1: Results of Lyapunov functions for Van der Pol oscillator. (a) Value of LNN 1. (b) Value of LNN 2, where the values of Lyapunov function in (a) are positive across the entire region, indicating that the proposed LNN structures satisfy the Lyapunov conditions. (c) Comparison of the stability regions for the actual nonlinear system obtained under different methods. The proposed method outperforms SOS 1, PF, and NLF due to a larger stability region.

### 5.2 LINEAR PATH FOLLOWING

Linear path following problem is a classic nonlinear system and is considered in this case study. Two LNN models are developed using the same structure as in Section 5.1, and the Lyapunov functions identified as SOS (Papachristodoulou & Prajna, 2005) and PF (Feng et al., 2024) are selected as benchmarks. Fig. 2 shows the satisfaction of Lyapunov conditions, and the effectiveness of the proposed method with a larger stability region compared to the PF and SOS methods. Detailed experimental results for the linear path following problem are provided in Appendix D.

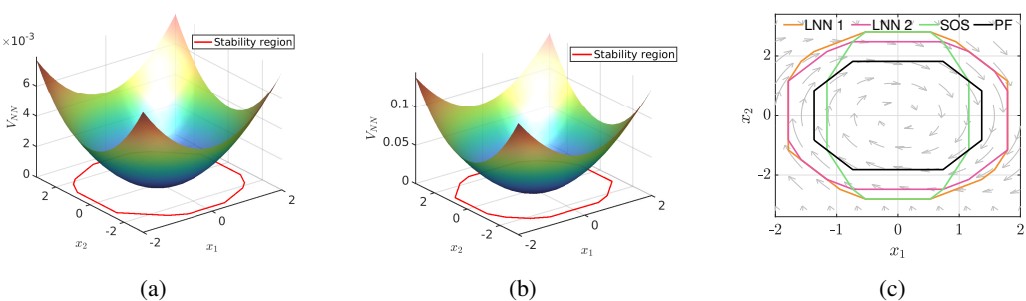

Figure 2: Results of Lyapunov functions for the linear path following problem. (a) Value of LNN 1. (b) Value of LNN 2. (c) Comparison of the stability regions obtained under different methods.

### 5.3 CHEMICAL REACTOR EXAMPLE

The last case study uses a chemical process example to demonstrate the effectiveness of the proposed data-driven framework to design the Lyapunov function and identify the stability region. Specifically, a well-mixed and non-isothermal continuous stirred tank reactor (CSTR) is considered, which contains two state variables $C_A$ (the concentration of the reactant in the reactor) and $T$ (the reactor temperature). The objective is to identify the largest stability region of the nonlinear chemical reactor where the first-principles model is unknown. The PF identified in Wu et al. (2019) and the NLF method in Chang et al. (2019); Min et al. (2023) are selected as benchmark models. An FNN model is first developed to capture the dynamics of the nonlinear system. Sufficient data are collected from the nonlinear system to ensure that a small approximation error can be guaranteed. Then, an LNN is constructed with the structure and loss function introduced in Section 3, and the optimal LNN for the FNN model is identified by solving a Bayesian optimization problem. Detailed experimental results, including the description of CSTR, the development of FNN and LNN models, and the optimization process, can be found in Appendix E.

Fig. 3 illustrates the optimization process over five iterations, showing the largest stability region obtained for five different LNN functions. The stability region progressively expands with each iteration, increasing the number of stable state samples from 42 to 1479, ultimately leading to the identification of the optimal Lyapunov function. Note that conventional methods in Chang et al. (2019); Zhou et al. (2022); Min et al. (2023) to design the LNN perform poorly on complex nonlinear systems such as the CSTR. Specifically, the stability region identified under the Control with Inherent Lyapunov stability (CoILS) method contains only 35 state samples, despite all samples satisfying $V > 0$ and $\dot{V} < 0$. The methods to identify the counterexamples (CEs) that violate the requirements for the Lyapunov function and then add new samples into the training dataset are unlikely to converge without structure constraints as the complexity of the nonlinear dynamics increases (Feng et al., 2024). Fig. 4 shows that both the percentage of counterexamples in the training dataset and the training loss under the NLF oscillate over 70 epochs, indicating the drawbacks (even the collapse) of the conventional methods. Due to the structured design of the LNN model, all states maintain positive Lyapunov values. The value of the loss function decreases throughout the training process, and the number of CEs is reduced and converges to a small value, which illustrates the effectiveness of the developed LNN structure in satisfying Lyapunov conditions. Note that the weight parameters in the LNN are initialized deterministically based on the optimized variables (i.e., the predefined boundaries of the neural network weights). As a result, repeated runs produce identical results and error bars are not applicable.

To further evaluate the effectiveness of the stability region for the underlying nonlinear system, the Lyapunov function and neural network-based controller are applied to the reactor system. Specifically, the optimal solution contains $1,479$ samples within the stability region $V_{nn} \leq 26.0$ under the condition $\dot{V}_{nn}(F_{NN}, \Phi_{NN}) + 2V_{nn} < 0$, where $\dot{V}_{nn}(F_{NN}, \Phi_{NN}) = \frac{\partial V_{nn}(x)}{\partial x} F_{nn}(x, \Phi_{nn}(x))$. The results in Fig. 3f show that 6,385 out of 6,400 state samples satisfy $\dot{V}_{nn}(F, \Phi_{NN}) \leq 0$. The unstable points for the actual nonlinear system are all located outside the identified stability region. The optimal LNN outperforms the PF (Wu et al., 2019), of which the stability region is with $1,126$ samples.

These simulation results validate the proposed data-driven framework's effectiveness in designing the Lyapunov function and identifying a stability region with strong generalization performance.

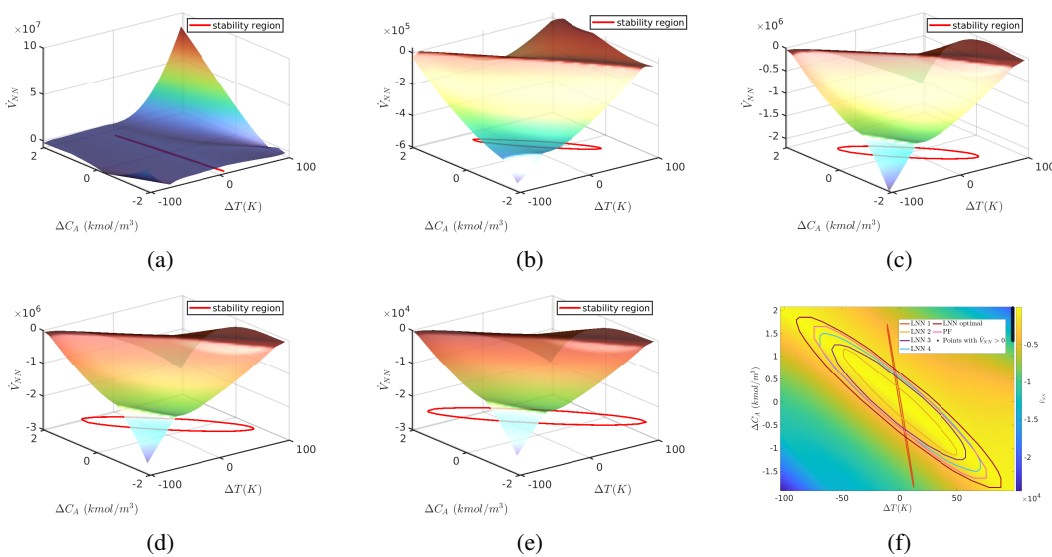

Figure 3: Results of Lyapunov functions for the CSTR. (a), (b), (c), (d), and (e) Values of LNNs identified over 5 iterations during the optimization process. (f) Comparison of stability regions identified by LNNs and PF. The stability region expands with each iteration, and the optimal LNN outperforms the PF method with a larger stability region.

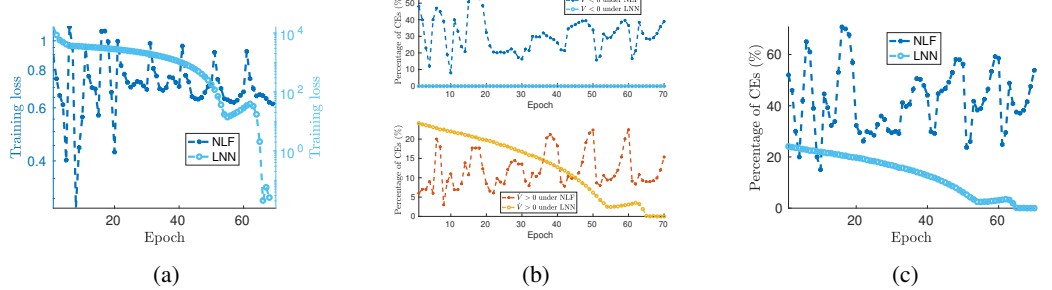

Figure 4: Comparison of the proposed LNN method and the NLF method in Chang et al. (2019); Zhou et al. (2022). (a) Training performance of LNN and NLF. (b) Number of counterexamples (CEs) that violate the Lyapunov conditions $V_{NN} > 0$ (top) and $\dot{V}_{NN} < 0$ (bottom). (c) Total number of CEs observed throughout the training performance.

## 6   CONCLUSIONS

This work proposed a novel LNN framework for nonlinear systems with unknown dynamics. The LNN was designed to satisfy control Lyapunov function conditions while maximizing the stability region. An optimization problem was formulated to identify the optimal LNN, with its training process embedded to enforce Lyapunov conditions over sampled states. Theoretical guarantees were provided to ensure generalization beyond the training set, and the identified stability region remained valid despite model approximation errors. Simulations of various nonlinear systems demonstrated the effectiveness of the proposed machine learning approach in designing Lyapunov functions and accurately characterizing stability regions.

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

## REPRODUCIBILITY STATEMENT AND LLM USAGE STATEMENT

We provide implementation code, experimental scripts, dataset descriptions, and complete theoretical analyses and proofs in the supplementary materials to ensure reproducibility. We used a large language model (LLM) for proofreading and polishing. All authors reviewed and edited the content to ensure scientific accuracy.

## A  PROOF DETAILS

### A.1  PROOF OF THEOREM 1

The proof of Theorem 1 is shown as follows.

*Proof.* First, we can obtain the lower bound of $V_{nn}$ using Eq. 6c and Eq. 6d as

$$V_{nn} = o(\sum_{j=1}^{m} w_{oj} h_j(\sum_{i=1}^{n} w_{ji} x_i)) + \phi(x) \geq \phi(x) \geq k_5 \|x\|^2 \tag{16}$$

The upper bound of $V_{nn}$ can be calculated as

$$V_{nn} \leq k_3 (\sum_{j=1}^{m} w_{oj} h_j(\sum_{i=1}^{n} w_{ji} x_i))^2 + k_6 \|x\|^2$$

$$\leq k_3 (k_1 \sum_{j=1}^{m} (w_{oj} \sum_{i=1}^{n} w_{ji}) |x_i|)^2 + k_6 \|x\|^2$$

$$\leq k_3 (mn k_1 U B_w^2 \sum_{i=1}^{n} |x_i|)^2 + k_6 \|x\|^2 \tag{17}$$

$$\leq n k_3 (mn k_1 U B_w^2)^2 \|x\|^2 + k_6 \|x\|^2$$

where the first line is derived using the characteristic of $o(x)$ in Eq. 6c and the requirement of $\phi(x)$ in Eq. 6d. The second line is obtained using Eq. 6b. The upper bound of the weight parameters defined in Eq. 6a derives the third line. The final line is obtained using the definition of the Euclidean norm.

Then, we can calculate the partial derivative of $\mathbf{o}(x)$ as

$$\frac{\partial \mathbf{o}(x)}{\partial x_i} = o^{'}(\sum_{j=1}^{m} w_{oj} h_j(\sum_{i=1}^{n} w_{ji} x_i)) \times \left( \sum_{j=1}^{m} (w_{oj} w_{ji} h_j^{'}(\sum_{j=1}^{n} w_{ji} x_i)) \right)$$

$$\leq k_4 k_1 \sum_{j=1}^{m} (w_{oj} \sum_{i=1}^{n} w_{ji}) |x_i| \times \sum_{j=1}^{m} (w_{oj} w_{ji} k_2) \tag{18}$$

$$\leq k_1 k_2 k_4 nm^2 U B_w^4 \sum_{i=1}^{n} |x_i|$$

where the second line is derived using the characteristic of the derivative of activation functions in the hidden and output layers.

Using Eq. 18, we have

$$\left\| \frac{\partial V_{nn}(x)}{\partial x} \right\| \leq \sqrt{(\frac{\partial o(x)}{\partial x_1})^2 + \dots (\frac{\partial o(x)}{\partial x_i})^2 + \dots (\frac{\partial o(x)}{\partial x_n})^2} + \left| \frac{\mathrm{d}\phi(x)}{\mathrm{d}x} \right|$$

$$\leq k_1 k_2 k_4 m^2 n^2 U B_w^4 \sum_{i=1}^{n} |x_i| + k_7 \|x\| \tag{19}$$

$$\leq \left( k_1 k_2 k_4 m^2 n^2 U B_w^4 \sqrt{n} + k_7 \right) \|x\|$$

The proof is completed using Eq. 16, Eq. 17, and Eq. 19. □

## A.2 PROOF OF THEOREM 2

The proof of Theorem 2 is shown as follows.

*Proof.* For $x \in \mathcal{B}_r(x_p, r)$, we select the controller candidate as $\Phi(x_p)$. Then, using the characteristic of the nonlinear function $F(x, u)$ designed in Eq. 3, we have the following inequality that holds $\forall x \in \mathcal{B}_r(x_p, r)$:

$$
\begin{aligned}
\frac{\partial V_{nn}(x)}{\partial x} F(x, \Phi(x)) &= \frac{\partial V_{nn}(x_p)}{\partial x_p} F(x_p, \Phi(x_p)) + \frac{\partial V_{nn}(x)}{\partial x} F(x, \Phi(x)) - \frac{\partial V_{nn}(x_p)}{\partial x_p} F(x_p, \Phi(x_p)) \\
&\leq -c_a \|x_p\|^2 + L_2 r \\
&\leq -c_a \|x_p\|^2 + (L_2 - c_b) \frac{c_a - c_b}{L_2 - c_b} \|x_p\|^2 + c_b \frac{c_a - c_b}{L_2 - c_b} \|x_p\|^2 \\
&\leq -c_b \|x\|^2
\end{aligned}
$$
(20)

where the second line is derived with the Lipschitz constant for $\frac{\partial V_{nn}(x)}{\partial x} F(x, \Phi(x))$, and the third line is obtained using the definition of $r$. $\qquad\square$

## A.3 PROOF OF THEOREM 3

The proof of Theorem 3 is shown as follows.

*Proof.* We can obtain

$$
\begin{aligned}
\frac{\partial V_{nn}(x)}{\partial x} F(x, \Phi_{nn}(x)) &= \frac{\partial V_{nn}(x)}{\partial x} F_{nn}(x, \Phi_{nn}(x)) + \frac{\partial V_{nn}(x)}{\partial x} (F(x, \Phi_{nn}(x)) - F_{nn}(x, \Phi_{nn}(x))) \\
&\leq -\lambda V_{nn}(x) + C_\lambda \|x\| \left\| \frac{\partial V_{nn}(x)}{\partial x} \right\| \\
&\leq (-\lambda k_5 + C_\lambda P_V) \|x\|^2 \\
&\leq \frac{-\lambda k_5 + C_\lambda P_V}{2k_3 (mnk_1 U B_w^2)^2 + k_6} V_{nn}(x)
\end{aligned}
$$
(21)

where the second line is obtained using Eq. 12, and the third line is derived using Eq. 19. $\qquad\square$

# B DISCUSSION ON THEOREMS AND TRADITIONAL METHODS

## B.1 DISCUSSION ON THEOREM 1

Note that Theorem 1 also holds for networks with multiple hidden layers, as the following inequality can be derived for $h(t)$ satisfying Eq. 6a and Eq. 6b:

$$
h(h(t)) \leq k_1^2 |t|, 0 < \frac{\mathrm{d} h(h(t))}{\mathrm{d} t} \leq k_2^2
$$
(22)

Following a similar process, the requirements for the Lyapunov function in Theorem 1 can also be satisfied for the LNN model with multiple hidden layers.

*Remark* 2. The adaptation term $\phi(x)$ satisfies the requirements in Eq. 2, and can be selected as a Lyapunov function on its own. It can be considered a candidate for the Lyapunov function, although its performance may be suboptimal, potentially yielding a conservative stability region. It is designed to ensure that $c_1 \|x\|^2 \leq V_{nn}(x)$ and balance the shape of the stability region derived. In addition, the form of $\phi(x)$ can be chosen either as a fixed function or as neural network layers, providing flexibility for the designed LNN to accommodate different scenarios.

### B.2  DISCUSSION ON THEOREM 3

Unlike the standard assumption that the approximation error $\|v\|$ for the FNN model is bounded by a constant, this manuscript assumes that the error bound between the predicted and actual states is dependent on $\|x\|$. This assumption is crucial for the Lyapunov function design and can be enforced by selecting an appropriate loss function for neural network training. For example, the mean absolute percentage error (MAPE) (De Myttenaere et al., 2016), which measures the relative difference between the predicted and actual values, serves as a suitable choice to satisfy the assumption of Theorem 3.

*Remark* 3. In this manuscript, the system dynamics is assumed to be unknown and the controller $\Phi_{nn}(x)$ is designed based on a neural network model. However, specific information for the system dynamics may be required for the design of $\Phi_{nn}(x)$. For example, in control-affine nonlinear systems under the Sontag controller, the values of the nonlinear functions within the system are necessary for controller design. In such cases, the FNN model can be designed as $F_{nn}(x) = f_{nn}(x) + g_{nn}(x)u$ in control-affine form, allowing the estimation of both the state vector and the underlying nonlinear functions.

### B.3  DISCUSSION WITH TRADITIONAL METHODS

The SOS method constructs a Lyapunov function by enforcing $V(x)$ and $-\dot{V}(x)$ to be sums of squares, ensuring positivity and semi-positivity (Papachristodoulou & Prajna, 2002; 2005; Peet & Papachristodoulou, 2012; August & Barahona, 2022). Its main limitation is the restriction to polynomial forms, which reduces representational capacity for non-polynomial functions (e.g., $1/x$, $\sin(x)$, and $e^x$), often yielding conservative stability regions. The stability region depends not only on samples satisfying Lyapunov conditions but also on the functional form of $V$, since it is defined by the largest level set of $V$ contained within the valid state set. The proposed framework overcomes these limitations by customizing neural network structures to satisfy Lyapunov requirements (Eq. 2), formulating an optimization problem to maximize the stability region, and leveraging diverse activation functions (Theorem 1) beyond the polynomial constraint of SOS.

Solving Zubov's PDE can exactly characterize the ROA but is computationally challenging. As mentioned in Meng et al. (2025), constructing a Lyapunov function that reduces ROA conservativeness is difficult for the approach, especially without known dynamics. While Zubov's method offers exact ROA for accurate models, its PDE complexity limits practicality.

## C  EXPERIMENT: APPLICATION TO THE VAN DER POL OSCILLATOR

The dynamics of the Van Der Pol oscillator can be described by the following equation:

$$\begin{aligned}
\dot{x}_1 &= x_2 \\
\dot{x}_2 &= -x_1 + (x_1^2 - 1)x_2
\end{aligned} \tag{23}$$

The objective is to identify the optimal LNN with the largest stability region. Specifically, the search space is defined as $\{(x_1, x_2) \in \mathbb{R}^2 \mid |x_1| \leq 2, \; |x_2| \leq 2\}$. The Lyapunov neural network consists of one hidden layer of four neurons, one output layer with one neuron, and two adaptation layers with one neuron each. The hidden layer uses the $\tanh(x)$ activation function, while the output and adaptation layers employ the quadratic activation $x^2$. The number of samples within the stability region identified under different Lyapunov functions is summarized in Table 1. To develop LNN 2, a feedforward neural network (FNN) is first trained to approximate the system dynamics. Based on this model, LNN 2 identifies the largest stability region for the FNN approximation, which also holds validity for the actual nonlinear system.

Table 1: Number of samples within the stability region identified for the Van der Pol oscillator

| Lyapunov function | LNN 1 | LNN 2 | SOS 1 | SOS 2 | PF | NLF | Valid region |
|---|---|---|---|---|---|---|---|
| Number of samples | **160** | **154** | 34 | **180** | 88 | 61 | 400 |

The time derivatives of LNN 1 and LNN 2 are shown in Fig 5. In this work, the Lyapunov function is represented via the neural network (LNN). Since the structure and activation functions of LNNs

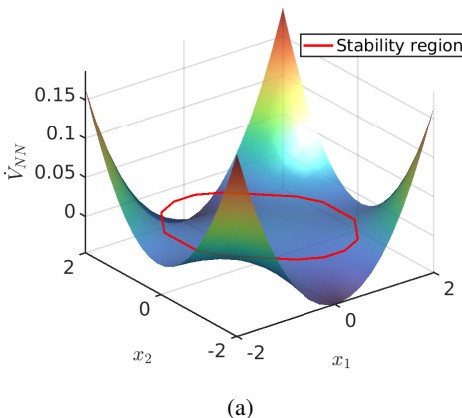 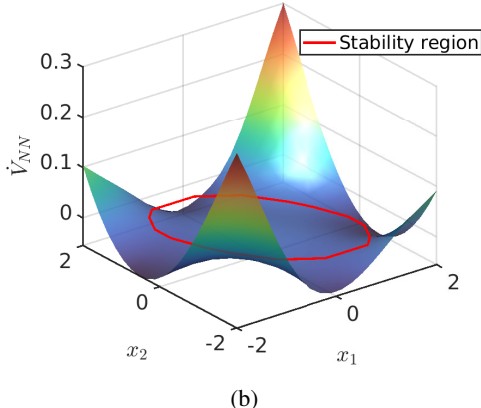

(a)                                                    (b)

Figure 5: Results of Lyapunov functions for the Van der Pol oscillator problem. (a) Time derivative of LNN 1. (b) Time derivative of LNN 2.

are predefined, and the weight parameters can be obtained via the Pytorch function , the explicit form of the Lyapunov function can be derived from Eq. 5 if needed, which gives the analytical expression of the LNN output (i.e., the value of the Lyapunov function) with respect to the input (i.e., the state vector). The Lyapunov function LNN 1 and LNN 2 can be represented as $V_{NN} = (W_2 tanh(W_1 x))^2 + B_1 x_1^2 + B_2 x_2^2$. Specifically, the weight parameters for LNN 1 are:

$$W_1 = \begin{bmatrix} 0.2942 & 0.3475 & 0.6243 & 0.2099 \\ 0.0742 & 0.0742 & 0.0742 & 0.0742 \end{bmatrix}^T, \quad W_2 = [0.0742 \quad 0.0742 \quad 0.0742 \quad 0.0742],$$

$$B_1 = [0.0245], \quad B_2 = [0.0879]$$

*Remark* 4. We conducted additional experiments using Monte Carlo (MC) sampling to estimate the volume of the stability region. Specifically, a large number of states were uniformly sampled within the region of interest, and the fraction of samples satisfying the Lyapunov stability conditions was used to approximate the region's volume. The estimated volumes obtained via MC closely matched those from grid-based methods, indicating that our findings are robust to the choice of sampling strategy. For example, in Van der Pol oscillator, the search space is $(x_1, x_2) \in \mathbb{R}^2$, $|x_1| \leq 2$ and $|x_2| \leq 2$, representing a square of area 16. Using grid search, we identified a stability region containing 160 out of 400 total samples. When applying MC with the same number of points, the estimated volume was also 6.4. Moreover, we evaluated the impact of sampling resolution by varying the number of points from 40 to 400 and 4000. Across all resolutions, the estimated volumes remained stable (within 5% variation), further confirming the robustness of our stability region estimates.

## D  EXPERIMENT: APPLICATION TO THE LINEAR PATH FOLLOWING

The dynamics of the linear path following system can be represented as:

$$\dot{x}_1 = a sin(x_2)$$
$$\dot{x}_2 = -x_2 - ab \frac{sin(x_2)}{x_2} x_1 \tag{24}$$

where $x_1$ and $x_2$ denote the distance and angel error between the robot trajectory and the reference signal. $a = 2$ is a constant and $b = 6 \; ms^{-1}$ is the constant velocity. The valid space for the design of the Lyapunov function is selected as $\{(x_1, x_2) \in \mathbb{R}^2 \mid |x_1| \leq 2, \; |x_2| \leq \pi\}$ (Feng et al., 2024). We list the number of samples within the stability region identified by different methods in Table 2.

The feedforward neural network (FNN) is developed to model the system dynamics. The time derivatives of LNN 1 and LNN 2 are shown in Fig 6. The Lyapunov function LNN 1 and LNN 2 can be represented as $V_{NN} = (W_2 tanh(W_1 x))^2 + B_1 x_1^2 + B_2 x_2^2$.

Table 2: Number of samples within the stability regions identified for the path following system

| Lyapunov function | LNN 1 | LNN 2 | SOS | PF | Valid region |
|---|---|---|---|---|---|
| Number of samples | **264** | **244** | 180 | 144 | 400 |

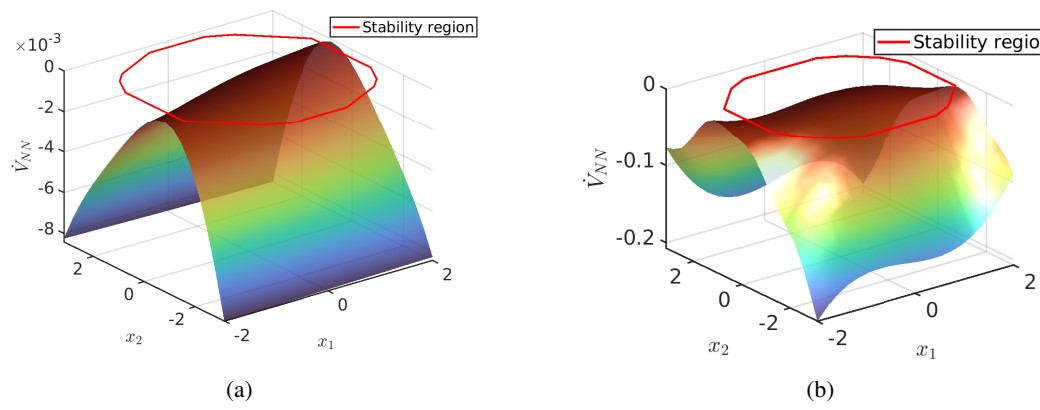

(a)  (b)

Figure 6: Results of Lyapunov functions for the linear path following problem. (a) Time derivative of LNN 1. (b) Time derivative of LNN 2.

## E  EXPERIMENT: APPLICATION TO A CSTR PROCESS EXAMPLE

The dynamics of the CSTR process can be described as follows:

$$
\begin{aligned}
\frac{dC_A}{dt} &= \frac{F}{V}(C_{A0} - C_A) - k_0 e^{\frac{-E}{RT}} C_A^2 \\
\frac{dT}{dt} &= \frac{F}{V}(T_0 - T) - \frac{\Delta H}{\rho_L C_p} k_0 e^{\frac{-E}{RT}} C_A^2 + \frac{Q}{\rho_L C_p V}
\end{aligned}
\tag{25}
$$

where $C_A$ and $T$ denote the concentration of the reactant and the reactor temperature, respectively. $Q$ is the rate of heat supply, and $C_{A0}$ denotes the concentration of the reactant in the feed stream. The state variable of the nonlinear system is $[C_A, T]$, and the manipulated input is $[Q, C_{A0}]$. The CSTR process is illustrated in Fig. 7. The number of samples within the stability region identified under different Lyapunov functions is listed in Table 3.

The identified optimal LNN can be described as $V_{NN} = (W_2(W_1 x))^2 + B_1 x_1^2 + B_2 x_2^2$, and the weight parameters are:

$$
W_1 = \begin{bmatrix} 6.5721 & 6.6834 & 6.7935 & 5.5942 & 6.6504 & 6.6166 & 6.6155 & 6.0774 \\ 0.1282 & 0.1282 & 0.1282 & 0.1282 & 0.1282 & 0.1282 & 0.1282 & 0.1282 \end{bmatrix}^T,
$$

$$
W_2 = \begin{bmatrix} 0.1505 & 0.1597 & 0.1678 & 0.1282 & 0.1566 & 0.1535 & 0.1369 & 0.1282 \end{bmatrix},
$$

$$
B_1 = [0.3678], \quad B_2 = [0.0564]
$$

Table 3: Number of samples within the stability regions identified for CSTR

| Lyapunov function | LNN 1 | LNN 2 | LNN 3 | LNN 4 | LNN optimal | PF | Valid region |
|---|---|---|---|---|---|---|---|
| Number of samples | 42 | 393 | 755 | 1018 | **1479** | 1126 | 6400 |

### E.1  DEVELOP OF FNN FOR THE SYSTEM DYNAMICS

The unstable steady-state of the CSTR process is
$[C_{As}, T_s, Q_s, C_{A0s}] = [1.95\ kmol/m^3, 402\ K, 0\ kJ/h, 4\ kmol/m^3]$, and the parameters of the

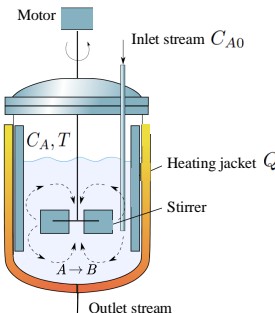

Figure 7: Illustration of CSTR process.

process are selected as (Wu et al., 2019). Specifically, the manipulated input is constrained by $|\Delta Q| \leq 5 \times 10^5 \ kJ/h$ and $|\Delta C_{A0}| \leq 3.5 \ kmol/m^3$, where $\Delta Q = Q - Q_s$ and $\Delta C_{A0} = C_{A0} - C_{A0s}$. A total of 160,000 samples are generated through open-loop simulations of Eq. 25, where the selection of initial states and the implementation of control signals follow the procedure outlined in Wu et al. (2019).

The valid region selected for the design of Lyapunov function is $\{(\Delta C_A, \Delta T) \in \mathbb{R}^2 \mid |\Delta C_A| \leq 1.9 \ kmol/m^3, \ |\Delta T| \leq 100 \ T\}$, where $\Delta C_A = C_A - C_{As}$, and $\Delta T = T - T_s$.

In the identification of the stability region, a neural network-based controller $\Phi_{nn}(x)$ is selected in the form of Sontag controllers. This controller ensures the stability of the closed-loop system in the absence of input constraints (Lin & Sontag, 1991). The values of the functions $f$ and $g$ in the control-affine nonlinear system are required to construct the Sontag controller. Thus, in addition to predicting the time derivative of the state, the neural network model must also predict the nonlinear functions. The neural network model is developed in control-affine form:

$$F_{nn}(x, u) = f_{nn}(x) + g_{nn}(x)u \tag{26}$$

where $f_{nn}$ and $g_{nn}$ denote the prediction of the nonlinear functions $f$ and $g$, respectively.

### E.2 DEVELOPMENT OF LNN AND NEURAL NETWORK-BASED CONTROLLER

To develop the LNN function, an FNN model is first constructed to satisfy the requirements in Eqs. 6a-6c. The input to the FNN model consists of the system states, while the output represents the Lyapunov function value. The bias parameters in the neural network model are set to 0, and the weight parameters are designed as bounded with the function "torch.nn.Parameter()". The activation function in the hidden layers is selected as $h(x) = x$, while the output layer uses $o(x) = x^2$ as the activation function, to meet Eqs. 6b-6c. In addition, new layers have been added for the adaptation terms, of which the weight parameters are fixed and not updated in the training process for simplicity. The new added layers are in the form of output layers with activation function $o^a(x) = k^a(x^a)^2$, where $x^a$ is the RNN input and $k^a$ denotes the fixed positive weight for the new layer.

The time derivative of the LNN model in the loss function is calculated using the neural network model in Eq. 26. A candidate controller is introduced for the calculation without accessing the exact first-principles model, which is designed in Sontag control law form. The $i$-th component of the candidate controller can be described as:

$$\tilde{\Phi}_{nn}^i(x) := -\frac{L_{f_{nn}}V_{nn} + \sqrt{L_{f_{nn}}V_{nn}^2 + \|L_{g_{nn}}V_{nn}\|^4}}{\|L_{g_{nn}}V_{nn}\|^2} L_{g_{nn}^i}V_{nn} \tag{27}$$

where $L_{f_{nn}}V_{nn} = \frac{\partial V_{nn}}{\partial x}f_{nn}$, and $L_{g_{nn}}V_{nn} = \frac{\partial V_{nn}}{\partial x}g_{nn}$, $L_{g_{nn}}V_{nn} = [L_{g_{nn}^1}V_{nn}, \ldots, L_{g_{nn}^m}V_{nn}]$. The values of $f_{nn}$ and $g_{nn}$ are predicted by the FNN model in the control-affine form, and the partial derivative of the $V_{nn}$ (i.e., the output of LNN) to the state (i.e., the input of LNN) is obtained

via the function "torch.autograd.grad()". If the value of $\Phi_{nn}^i$ falls within the allowable range of the manipulated input, it is directly implemented; otherwise, the corresponding input constraint is enforced. Specifically, the Sontag controller can render the time derivative of the Lyapunov function $V_{nn}$ negative over the entire state space, if there is no input constraint. In the development of the candidate controller $\Phi_{nn}$, the LNN and the FNN model for the nonlinear system are utilized. Moreover, the candidate controller can be applied to actual nonlinear systems, which also holds effective to develop the stability region.

### E.3 SELECTION OF OPTIMAL LNN VIA BAYESIAN OPTIMIZATION

The optimization problem in Eq. 14 presents significant complexity due to the max-max structure, the nonlinear dynamics, the calculation of the LNN model's time derivative, and the coupling between training and stability region identification. A key challenge lies in selecting the optimized variables. The boundaries of the weight parameters in the LNN model are chosen as the optimization variables. Specifically, $a$ is the lower bound of the weight in the hidden and output layers. $b$ and $c$ denote the fixed weight in the adaptation layer $\phi_x$ of Eq. 5 for the state variables $C_A$ and $T$, respectively. The maximum number of training epochs for the LNN model is set to 70. After each epoch, the number of states within the stability region is computed and recorded. The maximum number of stable states across all 70 epochs is selected as the final objective value, and the corresponding epoch is identified as the optimal training epoch for the LNN model under the given optimization parameters. The constraint for the optimized variables $[a, b, c]$ is set to $[1 \times 10^{-5}, 1]$. The maximum number of optimization iterations is 150. The optimization problem is implemented with the "BayesianOptimization" function in python package "bayes_opt" (N., 2014).

The optimal LNN model yields the largest stability region, containing 1,479 state samples, with optimized variables $[a, b, c] = [0.13, 0.37, 0.06]$. The calculation process for determining the number of samples within the stability region remains consistent across iterations. Here, we illustrate the computation of the objective function in Eq. 14 using the optimal set of parameters. In iteration 105, the set of optimized variables is selected as $[0.13, 0.37, 0.06]$ through the acquisition function, which defines the lower bound of the weight parameters for the training of the LNN model. Specifically, the lower bound of the weight parameters in the hidden and output layers is set as 0.13, while the fixed weights in the adaptation layers are chosen as 0.37 and 0.06 for the state $C_A$ and $T$, respectively. The partial derivative of the LNN to the state $\frac{\partial V_{nn}}{\partial x}$ is obtained using the Pytorch function "autograd.grad()". The training epoch is set as 70, and the Adam optimizer is chosen to update the weight parameters with the learning rate 0.1, which is commonly used in the neural network training process.

*Remark* 5. In this work, Gaussian process regression is the selected surrogate model in Bayesian optimization to model the objective function. Given a set of observations, the Gaussian process is trained to model the posterior distribution of the objective as $f(x) \sim N(\mu(x), \sigma^2(x))$, where $\mu(x)$ and $\sigma^2(x)$ are the mean and variance, respectively. The expected improvement function is a commonly used acquisition function in Bayesian optimization, which estimates the expected improvement over the current best objective value predicted by the surrogate function, to update the optimized variables in each iteration. Specifically, the expected improvement at the $n$-th iteration is defined as $EI_n(x) := \mathbb{E}_n[max(f(x) - f_n^a), 0]$, where $f(x)$ is modeled by the surrogate (Gaussian process), and $f_n^a$ denotes the best objective value observed so far. In this work, the Bayesian optimization package (N., 2014) is utilized to perform the optimization process for the identification of the optimal Lyapunov neural network. The optimized variable is the weight boundary for the LNN models. At each iteration, the LNN is trained under the given weight constraints over several epochs, and the number of samples within the identified stability region is used as the objective function value. Gaussian process regression models the relation between the weight boundaries and the corresponding objective function values, which is then used by the expected improvement acquisition function to update the optimized variable. The optimization process terminates when the objective value does not improve for $N$ iterations, where $N$ is selected based on each case.

*Remark* 6. During the training process, the number of states with a positive time derivative of the Lyapunov function is reduced, as guided by the loss function. However, an increased number of state points does not necessarily lead to a larger stability region, as the location of unstable points is unpredictable. Although the number of states with $\dot{V}_{nn} < 0$ increases, the stability region identified through the global search method exhibits significant oscillations. Even with many points showing a negative time derivative, unstable points near the origin of the state space can still disrupt the

construction of a large stability region. This performance not only underscores the importance of identifying the optimal epoch for training, but also highlights the complexity of the optimization problem. Consequently, the loss function alone is insufficient for building the largest stability region. The optimization problem must be updated to search for the best LNN model.

*Remark* 7. The comparison between the LNN model and the traditional polynomial Lyapunov function is as follows. The LNN model guarantees that the Lyapunov function remains nonnegative, whereas in the global search method for polynomial functions, this characteristic cannot be guaranteed. As the complexity of the polynomial function increases, it becomes challenging to verify whether the polynomial satisfies the requirements defined in Eq. 2. Moreover, while the parameters of the polynomial function are fixed, in the LNN model, the weight parameters are updated through a specific loss function designed to reduce the number of states with $\dot{V}_{nn} > 0$, which is more effective than the polynomial function.

*Remark* 8. In this approach, a region of the state space is first identified where the time derivative of the Lyapunov function $V_{nn}$ is rendered negative for the FNN model under the neural network-based controller $\Phi_{nn}$. Then, the largest level set of the Lyapunov function within the region is constructed as the stability region. Although the region is designed for the neural network model in Eq. 26, under specific modification (i.e., $\dot{V}_{nn} + 2V_{nn} < 0$), it is also valid for the actual nonlinear chemical processes described by Eq.25, as shown in Theorem 3.

*Remark* 9. The construction of the hypothesis function set and the selection of optimized variables can be more flexible. For example, $\mathcal{H}$ can include LNN models with varying numbers of hidden layers or neurons, and different activation functions can be chosen to develop the LNN model. The rich variety of the functions inside $\mathcal{H}$ may lead to a more effective LNN model, they also increase the complexity of solving the optimization problem. Consequently, new optimized variables must be carefully selected.

