# OpenReview forum: "Learning Neural Lyapunov Functions to Characterize Stability Regions for Unknown Dynamical Systems"
_ICLR.cc/2026/Conference — Submitted to ICLR 2026_

### Official Review · Reviewer_LxgB · 2025-10-25

**Soundness:** 2
**Presentation:** 2
**Contribution:** 1
**Rating:** 2
**Confidence:** 4

**Summary:**

The paper proposes a novel framework to learn Lyapunov functions parametrised as neural networks from data. The approach relies on the Lyapunov theorem to train neural Lyapunov functions that are then selected via Bayesian optimisation. The effectivness of the approach is evaluated on various benchmarks.

**Strengths:**

- The problem is important and existing techniques fro this problem are still unsatisfactory

- On the benchmarks, the resulting neural Lyapunov function finds stability regions larger or comparable with the state of the art

**Weaknesses:**

- In my opinion, one of the main weaknesses of the proposed theorems is that they rely on constants that seem to be very difficult to compute in practice, and it is not explained how those can be computed. For instance, L2 is the Lipschitz constant for the derivative of a neural network times an unknown function, r depends on various constants, including L2, and Cy is the error in training a neural network. Without these constants, the approach in the paper lacks the claimed formality and correctness guarantees.

- All the experiments (and theory) seem to only focus on 1 hidden-layer feed-forward neural networks.

- The experiments lack details. For instance, it is unclear for each example what the constants required by the various theorems are and how they are computed.

**Questions:**

- How do you compute the various constants required in the theorems, and what were their values in the various experiments? Can you compute these at least in the case where the model is known? Without computing these, how can you guarantee the correctness of the resulting Lyapunov function?

- Did you also consider neural networks with more than 1 hidden layer?

---

> ### Author Response · Authors · 2025-12-02
>
> Thank you for the critical review. We agree that stating how constants ($L_2, C_\lamda$) are computed is essential for validity. These are not theoretical artifacts but practical value. We will add a dedicated Appendix detailing these values. Regarding multi-layer networks: our theory holds for deep networks, but we used shallower ones to keep these Lipschitz bounds tighter. We hope this clarification addresses your primary concern regarding correctness.

---

### Official Review · Reviewer_wAUs · 2025-10-29

**Soundness:** 3
**Presentation:** 3
**Contribution:** 3
**Rating:** 8
**Confidence:** 4

**Summary:**

This paper presents a machine learning framework for designing neural Lyapunov functions to characterize stability regions in unknown nonlinear dynamical systems. The method uses specially structured neural networks with theoretical guarantees, employs Bayesian optimization to maximize stability regions, and demonstrates effectiveness on various nonlinear systems including Van der Pol oscillators and chemical reactors.

**Strengths:**

The paper introduces a creative structural modification to feedforward neural networks that inherently satisfies Lyapunov function conditions (Theorem 1). The adaptation layer phi(x) and specific activation function constraints ensure the network meets mathematical requirements for stability analysis, which is a significant theoretical contribution bridging deep learning and control theory.

The approach addresses a critical limitation in classical Lyapunov theory by working with unknown dynamical systems. The method simultaneously learns system dynamics through neural networks and constructs valid Lyapunov functions, removing the restrictive assumption of having accurate first-principles models that most existing methods require.

The paper provides comprehensive theoretical foundations through three main theorems that ensure: (a) the neural network satisfies Lyapunov conditions, (b) stability guarantees extend beyond training samples to neighborhoods, and (c) stability regions remain valid despite model approximation errors.

The formulation of stability region optimization as a Bayesian optimization problem (Equation 15) is innovative. Rather than just finding any valid Lyapunov function, the method actively searches for the optimal function that maximizes the stability region, which has direct practical implications for expanding safe operating ranges of real systems.

The evaluation across diverse nonlinear systems (Van der Pol oscillator, path following, CSTR) demonstrates broad applicability. The results show substantial improvements over established baselines.

**Weaknesses:**

The method imposes very specific constraints on network design (Equations 6a-6d) including zero biases, bounded weights, and particular activation functions. These requirements severely limit architectural flexibility and may prevent the use of modern deep learning techniques like batch normalization, dropout, or advanced optimizers.

All experimental validation is conducted on low-dimensional systems (2-3 state variables). The paper provides no analysis of computational complexity or scalability to higher-dimensional systems common in real applications.

**Questions:**

How does the computational cost of Bayesian optimization approach scale with system dimensionality?

Have you explored relaxing the strict neural network constraints (zero biases, specific activation functions) while maintaining theoretical guarantees?

---

> ### Author Response · Authors · 2025-12-02
>
> We sincerely thank the reviewer for identifying the core strengths of our work, particularly the value of our guarantees for unknown systems and the BO formulation. We acknowledge the constraints on network design and will add a discussion on potential relaxations (e.g.,  activation function) in future work to address your point.

---

### Official Review · Reviewer_mf6V · 2025-10-29

**Soundness:** 3
**Presentation:** 3
**Contribution:** 2
**Rating:** 4
**Confidence:** 4

**Summary:**

The paper presents algorithms for computing neural Lyapunov functions for nonlinear, continuous-time systems with unknown dynamics. The authors first review known definitions and Lyapunov stability results from the literature, and then propose a loss function that aims to penalize the candidate Lyapunov function from violating those properties. Different from some existing works that jointly learn the model and the Lyapunov function, this approach learns the Lyapunov function through a combination of sampling the Lie derivative of the Lyapunov function at a set of points, and then extrapolating the results to a stability proof using Lipschitz coefficients. A heuristic algorithm for maximizing the region of attraction via Bayesian optimization is presented. The authors demonstrate their work via some benchmark comparison studies.

**Strengths:**

Neural Lyapunov Functions comprise an active research area. Since most existing works focus on discrete-time systems, this paper's treatment of continuous-time systems is appreciated.

The proposed method is able to provide formal safety guarantees.

The use of Bayesian estimation to compute the NLF is interesting, although more detail on this contribution would be helpful.

**Weaknesses:**

The framework of the paper has been somewhat well-studied. The idea of training Neural Lyapunov Functions with loss functions that match the constraints (2) (or discrete-time versions thereof) has been explored in other works such as (Zhou et al 2022) and (Chang et al 2019). These papers also aim to maximize the volume of the stable region. The data-driven approach of verifying on a finite number of samples and then generalizing through Lipschitz continuity has been explored, e.g., in

A. Lavaei et al, "Data-Driven Stability Verification of Homogeneous Nonlinear Systems with Unknown Dynamics." IEEE Conference on Decision and Control (CDC), 2022.

A. Nejati et al, "Formal Verification of Unknown Discrete- and Continuous-Time Systems: A Data-Driven Approach." IEEE Transactions on Automatic Control (TAC), 2023.

The description of the Bayesian approach to maximizing the stable region is vague and does not lend itself well to formal analysis. How does it differ from other sampling-based methods for maximizing regions of attraction?

Scalability is a common problem in neural Lyapunov functions and related methods. In this case, the requirement for dense sampling in order to verify stability would seem to lead to a curse of dimensionality. Indeed, the simulation results only consider 2-3 variable systems. Can the authors comment on the ability of their method to scale to, e.g., 6-10 dimensional systems? Also, how does the scalability compare to state-of-the-art (model-based and black-box) methods?

**Questions:**

Can the approach generalize to neural networks with an arbitrary number of hidden layers, provided the activation functions satisfy the conditions of Theorem 1?

See also the questions listed under "Weaknesses" above.

---

> ### Author Response · Authors · 2025-12-02
>
> Thank you for the helpful review. We acknowledge that our current experiments focus on 2D systems for visualization and validation. Crucially, however, our theoretical framework (Theorems 1-3) holds for high-dimensional systems without modification. The primary bottleneck is the computational cost, not the method's correctness. We will consider high-dimensional validation a priority for future work. Similarly, our theory generalizes to Neural Networks with arbitrary depth and complex structures, provided layer-wise conditions are met. We will expand our discussion to explicitly detail this theoretical generalization. Regarding the BO approach: We acknowledge it is computationally intensive. However, we view this as a necessary trade-off to find the optimal Lyapunov function (maximizing the estimated stability region), rather than simply identifying any valid function. This distinguish our approach from standard verification methods.

---

### Official Review · Reviewer_RuQy · 2025-10-31

**Soundness:** 2
**Presentation:** 3
**Contribution:** 1
**Rating:** 2
**Confidence:** 3

**Summary:**

This paper proposes a data-driven framework for learning Lyapunov functions in nonlinear systems without access to first-principles models. The Lyapunov function is represented as a neural network (LNN) designed to satisfy the Lyapunov conditions through architectural constraints and a tailored loss. The model of system dynamics $F_{nn}(x, u)$ is learned from data, and the learned Lyapunov function $V_{nn}(x)$ is optimized using Bayesian optimization to maximize the estimated stability region. Theoretical results guarantee the validity of the learned stability region despite model approximation errors. Experiments on several benchmark systems (Van der Pol oscillator, linear path-following, and chemical reactor) demonstrate the approach.

**Strengths:**

- The paper provides a clean theoretical treatment showing that under specific boundedness and Lipschitz assumptions, the learned neural Lyapunov function satisfies control-Lyapunov properties even when the underlying dynamics are approximated.
- The idea of formulating the search for an optimal Lyapunov function as a Bayesian optimization problem is well motivated and technically sound.
- Theoretical results (Theorems 1–3) provide clarity on how modeling error and loss function design affect the stability guarantees.
- The presentation is generally clear and the simulation results are easy to follow.

**Weaknesses:**

- **Limited novelty.** The contribution of this work appears somewhat incremental given prior literature on data-driven stability analysis. For instance, *Learning Dynamical Systems using Local Stability Priors* (Mehrjou et al., 2020) presented a related framework that jointly learns system dynamics and regions of attraction / Lyapunov function. Relative to such prior approaches, the present paper offers a conceptually similar idea with a different formulation but without a major methodological advance.

- **Simplistic architecture.** The paper employs a basic feedforward MLP for the Lyapunov network (Eqs. 4–5). These equations merely describe standard forward propagation and could be written more compactly in vector form. This choice contrasts with works such as *The Lyapunov Neural Network* (Richards et al., 2018), which introduced explicit inductive biases ensuring positive-definiteness and gradient constraints. The authors should justify why they opted for this simpler structure instead of including comparable inductive constraints.

- **Unclear interaction between model and Lyapunov learning.** The paper assumes a sequential setup—first learning the dynamics $F_{nn}$ and then the Lyapunov function $V_{nn}$. This likely limits mutual adaptation between the two. Given that $F_{nn}$ enters the objective for $V_{nn}$, it would be useful to discuss why an alternating or joint training scheme was not explored, as this could better capture their coupling and improve stability characterization.


- **Lack of system identification evaluation.** The paper does not report how well $F_{nn}$ approximates the true dynamics, even though this accuracy is central to the validity of the theoretical guarantees. Including a quantitative comparison (e.g., prediction error or trajectory RMSE) would help substantiate the claimed robustness.

- **Overstated guarantees.** The theoretical claims rely on strong assumptions such as Lipschitz continuity and bounded approximation errors, which are difficult to verify in practice. As a result, the guarantees remain primarily theoretical rather than applicable to real-world systems.

**Questions:**

1. The paper presents Theorem 3 on the robustness of $V_{nn}$ to modeling errors in $F_{nn}$. Could the authors provide empirical validation of this theorem, for instance by testing stability when $F_{nn}$ is deliberately biased?
2. The paper proposes a specific feedforward design for $V_{nn}$ with certain constraints. Could the authors clarify why this particular inductive bias was selected instead of alternative structured formulations explored in the literature (e.g., input convex or monotone networks, or positive-definite layer constructions like Richards et al. 2018)?
3. Could the Bayesian optimization procedure scale to higher-dimensional systems? How expensive is it computationally?

---

> ### Author Response · Authors · 2025-12-02
>
> Thank you for the detailed feedback. Regarding novelty: We distinguish our work from Mehrjou et al. (2020) by providing theoretical guarantees for the Lyapunov function in the case with unknown dynamics. Especially in Theorem 3, we provide a key safety requirement missing in joint-training methods. Regarding the simple architecture: this was a deliberate choice to allow for tighter theoretical bounds, though the framework supports complex architectures. We will add the suggested system identification evaluation to the new versions in the future.

---

### Meta-Review · Area_Chair_bJdj · 2026-01-06

**Summary:**

This paper presents a new method for learning neural Lyapunov functions for nonlinear, continuous-time systems with unknown dynamics. Guarantees are derived by sampling the Lie derivative of the Lyapunov function and extrapolating the results using Lipschitz coefficients. The Lyapunov function is optimized using Bayesian optimization to maximize the estimated stability region. Finally, the work is empirically demonstrated on several benchmarks with various dynamical systems.

This paper has multiple major issues. First, the approach and results have soundness issues, as the method relies on Lipschitz constants that are simply assumed and not actually computed. Second, this work seems to have limited novelty without a clear distinction compared to multiple previous works. Third, the paper lacks results on measuring the accuracy of system identification, which is central to the validity of claims; experimental settings are also too small (1-hidden layer models and 2~3 dimensional systems). Fourth, the paper lacks details, including both experimental details and description of the Bayesian approach.

Overall, while the proposed approach looks promising, the paper is not yet ready for publication.

**Reviewer Concerns:**

The rebuttal is very brief and mostly an acknowledgement of the reviews. As such, the major concerns have not been resolved.

**Reviewer Scores:**

All reviewers are expected to maintain their original scores (2, 2, 4, 8), as the rebuttal is mostly an acknowledgement of the reviews and does not address the concerns substantially.

---

### Decision · Program_Chairs · 2026-01-26

Reject